# gUMI-BEAR, a modular, unsupervised population barcoding method to track variants and evolution at high resolution

**Shahar Rezenman**[1☯], **Maor Knafo**[1☯], **Ivgeni Tsigalnitski**[1], **Shiri Barad**[1], **Ghil Jona**[2], **Dikla Levi**[2], **Orly Dym**[3], **Ziv Reich**[1]*, **Ruti Kapon**[1]*

**1** Department of Biomolecular Sciences, Weizmann Institute of Science Rehovot, Rehovot, Israel, **2** Life Sciences Core Facilities, Weizmann Institute of Science Rehovot, Rehovot, Israel, **3** The Dana and Yossie Hollander Center for Structural Proteomics, Weizmann Institute of Science Rehovot, Rehovot, Israel

☯ These authors contributed equally to this work.
* ruti.kapon@weizmann.ac.il (RK); ziv.reich@weizmann.ac.il (ZR)

**Data Availability Statement:** All raw sequencing files are available from the ENA database Script for data analysis can be found in the rezenman/gUMI-BEAR github page

## Abstract

Cellular lineage tracking provides a means to observe population makeup at the clonal level, allowing exploration of heterogeneity, evolutionary and developmental processes and individual clones' relative fitness. It has thus contributed significantly to understanding microbial evolution, organ differentiation and cancer heterogeneity, among others. Its use, however, is limited because existing methods are highly specific, expensive, labour-intensive, and, critically, do not allow the repetition of experiments. To address these issues, we developed gUMI-BEAR (genomic Unique Molecular Identifier Barcoded Enriched Associated Regions), a modular, cost-effective method for tracking populations at high resolution. We first demonstrate the system's application and resolution by applying it to track tens of thousands of *Saccharomyces cerevisiae* lineages growing together under varying environmental conditions applied across multiple generations, revealing fitness differences and lineage-specific adaptations. Then, we demonstrate how gUMI-BEAR can be used to perform parallel screening of a huge number of randomly generated variants of the *Hsp82* gene. We further show how our method allows isolation of variants, even if their frequency in the population is low, thus enabling unsupervised identification of modifications that lead to a behaviour of interest.

## Introduction

With the advent of single-cell techniques, it became clear that significant heterogeneity exists within populations of seemingly identical cells [1–6]. However, how individual adaptations translate into success or failure, or how population dynamics are affected by individual variations and their degree, is not yet clear. A number of studies are beginning to shed light on these questions by employing procedures to quantify a population's clonal makeup [6–13]. Using these methods, multiple lineages within a population can be distinguished by the introduction of DNA barcodes, short segments of DNA inserted into cells' genomes and passed on to their progenies. By sequencing the barcodes, one can capture a snapshot of the clonal composition of a growing population at any given moment. Successive records can be used to trace

**Funding:** The author(s) received no specific funding for this work.

**Competing interests:** The authors have declared that no competing interests exist.

the prevalence of individual lineages through time [14], offering insight into adaptation dynamics and fitness changes as they occur *in the context of evolving populations* rather than in each strain on its own.

The study of heterogeneity need not be limited to individuals within a population but could also be considered in the context of a particular gene or pathway. Quantifying the effect of gene-variants allows mapping the fitness landscape of a gene towards the understanding of the biophysical and biochemical requirements of particular regions. Likewise, it can aid in designing disease treatments. Here, the conjugation of DNA barcodes to specific gene variants creates a one-to-one association between barcode and variant and allows to use barcode incidence as a proxy for the abundance of each variant in a population. Thus, the fitness of many variants can be tested together, in one experiment, in the evolutionary setting in which they compete with each other.

Cellular barcoding has thus become invaluable for understanding core evolutionary processes such as clonal interference [15], differentiation of organs [16] and heterogeneity development in cancer [1]. For variant testing, it has allowed the assessment of unprecedented numbers of variants [17,18]. Its utilisation is, however, limited because it is generally expensive, labour-intensive, highly specific to each biological system, and current modes of application do not lend themselves to repetition—a fundamental requirement in experimental research [19]. In addition, for variant testing, the use of barcoding is restricted to testing single modifications or short genomic regions in proximity to the barcode due to limitations on sequencing lengths.

Another ability lacking from current barcoding techniques, which prevents them from being used to study particular adaptations, is the ability to *directly* single out specific clones for further study. Currently, if particular clones are found of interest, based on the dynamics of the population, the procedure for distinguishing them for further study requires first separating single clones using other methods such as streaking or sorting and then sequencing the barcode of each colony or well to identify the variant. This procedure can be particularly cumbersome if one is looking for clones that are present at low frequencies, as the probability of finding a clone goes as its percentage in the population. Thus, a clone present below 1% requires (on average) more than 100 colonies to be sampled and sequenced, a feat that is both formidable and expensive.

To tackle some of the above limitations, we set out to simplify and streamline the barcoding process so that it can be applied to a broader range of studies by incorporating the following capabilities, (1) generation of identical library duplicates, (2) direct variant identification, (3) precision and modularity in the insertion point of the barcode and in the number of participating lineages and cells and, (4) potential for integration into different organisms. In this paper, we describe the method we devised to achieve these goals, which we named gUMI-BEAR (genomic Unique Molecular Identifier-Barcode Enriched Associated Regions) and show how it can be utilised to accomplish high-resolution lineage tracking at a relatively low cost as well as to directly test millions of variants of a gene in the context of a growing population. While it can be adapted to other systems, the setup we describe here is particularly applicable to asexual populations. We note that this method can be applied by any lab versed in standard molecular biology techniques and thus hope that it will be utilised by labs specialising in different areas.

## Results

### Building a barcoded library

gUMI-BEAR is based on the unsupervised barcoding of each cell in an initial population with a construct, the gUMI-box, that is inserted into the genome using CRISPR-Cas9. The

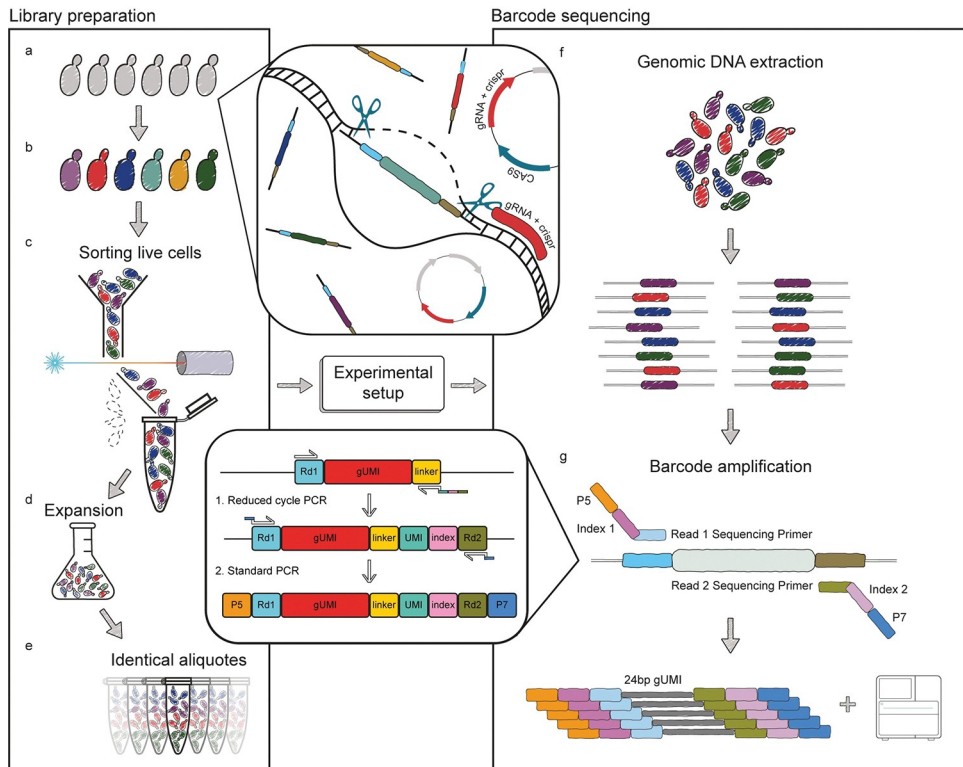

**Fig 1. Schematic illustration gUMI-BEAR.** An illustration demonstrating the method in its entirety, from library construction through experimental design, ending in the preparation of libraries for deep-sequencing. (**a, b**) An initial cellular population is barcoded by inserting unique 24 bp sequences into its genomes using CRISPR/Cas9 (centre top). (**c**) Following recovery from transformation, the desired number of live cells are sorted from the population, (**d**) are allowed to grow, and (**e**) the resulting population is divided into aliquots containing equal numbers of each barcoded lineage. The aliquots are subjected to the chosen experiment (centre middle), after which (**f**) the cells undergo DNA extraction. (**g**) A two-step polymerase chain reaction (PCR) targeting the barcode region (centre bottom) enables amplification of each lineage-associated variant (top) and subsequent sequencing of the barcode library (bottom).

construct includes a barcode, along with auxiliary elements required for successful transformation and ones that simplify downstream preparation of the samples for sequencing (see Fig 1 and Methods).

**Donor and gUMI box construction.** The barcode (gUMI) region of the gUMI-box we used contains a 24 bp random DNA segment preceded by five non-guanine nucleotides to allow efficient cluster generation by Illumina sequencers. This length of the random segment can be changed according to the requirements of each experiment, provided that the minimum of 25 base-pairs for cluster generation is kept. The barcode region is flanked by elements that direct the construct to its proper genomic locus upon transformation as well as components that shorten the preparation of samples for sequencing after the experiment (see Fig 1). Thus, upstream of the gUMI, we added a homology sequence for insertion into the proper location in the genome (Left Homology Arm, LHA) and the Illumina READ1 sequence that renders the segment ready for attachment of Illumina P5 adapters after the experiment [20]. Downstream of the gUMI we placed a generic linker sequence, to which READ2 will be attached following the experiment. The gUMI box is completed by a 20bp sequence that is complementary to a Hygromycin resistance cassette (Fig 2A).

The full donor that is transformed into yeast cells is composed of the gUMI-box, together with the hygromycin cassette under the TEF1 promoter, and the 3' homology sequence for

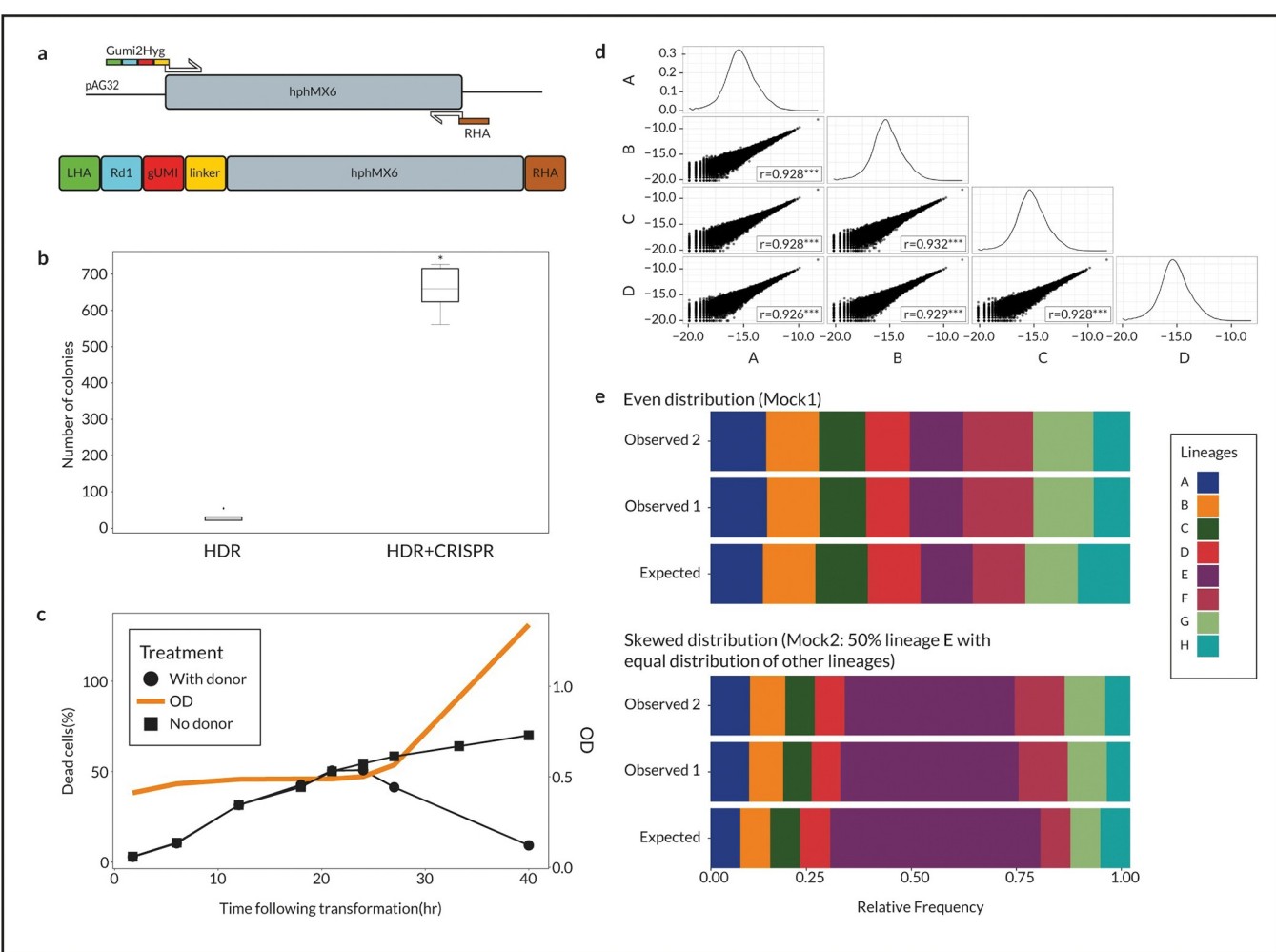

**Fig 2. Construction of the barcoded library. (a)** Schematic illustration of the donor DNA construction process **(b)** The number of colonies formed following transformation with donor DNA with (HDR+CRISPR) or without (HDR) a pCAS vector carrying the CRISPR/Cas9 machinery for five biological replicates of each treatment. Utilisation of CRISPR/Cas9 increased the number of colonies 20-fold (t-test between distributions, p = 3.89e-8). **(c)** Percentage of dead cells in a culture (left y-axis) as a function of time following transformation with (circles) and without (squares) donor DNA. Samples were taken at several time points to check membrane integrity using Propidium Iodide staining during the 40-h following transformation for quantification of the percentage of dead cells. The right y-axis presents optical density (OD; orange line) measurements for the culture showing that cells start to grow after 24 h **(d)** Four replicates of a library, A–D, were deep-sequenced to reveal their populations' barcode composition and the log-transformed frequencies of the lineages are plotted against each other. Pearson correlations were computed between all replicates and the Pearson coefficient, r, is presented at the bottom of each comparison. Plots on the diagonal present the log-transformed frequency distribution for each replicate (***$p < 0.0001$). **(e)** Bar plot of expected and observed lineage frequencies for two control populations, Mock1 (upper panel) and Mock2 (lower panel). Eight lineages were isolated from single colonies, and their barcodes were identified using Sanger sequencing. Their DNA was extracted and mixed in known ratios to create the Mock1 and Mock2 control populations. Each population was sequenced on two separate occasions, and the results underwent the same analysis pipeline to reveal its population composition. Each colour represents a single lineage according to the legend to the right of the graph.

genomic insertion (Right Homology Arm, RHA). It is formed by PCR amplification using the gUMI-box as the forward primer together with a reverse primer that consists a 20bp region complementary to the Hygromycin resistance cassette box together with the RHA sequence (Fig 2A).

As stated, we designed the gUMI-box to also aid the process of preparing the library for sequencing. We used the linker in a short-cycle PCR step to attach a UMI and a READ2 sequence to each amplicon (bottom centre of Fig 1). This endows each barcode that enters the analysis pipeline with a tag (Fig 1G) and prepares it for index attachment. Then, a second PCR

reaction is used to amplify the barcodes to the amount required for sequencing. The tag introduced in the first PCR provides a means to avoid biases that may be introduced in the multiple cycles required to produce enough DNA from small samples and thus increases the accuracy of lineage frequency quantification.

**Transformation, sorting and expansion.** To build a modular system that could potentially be applied to different organisms as well as to any genomic location, we decided to incorporate the gUMI using CRISPR double-strand break at the integration site [21,22] together with the native yeast homology-directed repair (HDR) system. Although, in yeast, HDR is usually sufficient for the integration of desired DNA into the genome by transformation, we found that the number of successful transformants increased 20-fold (Fig 2B) with the addition of the CRISPR system, which facilitates the preparation of larger libraries. Equally important, using CRISPR allowed us to easily change the locus of integration by simply replacing the gRNA in the pCAS vector and the homologous arms of the donor DNA while leaving the gUMI intact, thus satisfying our demand for flexibility in application.

To characterise the recovery of the barcoded population following transformation, we measured the optical density (OD) and the percentage of dead cells in the culture at several time points up to 40 hours post-transformation (see Methods). We found that after 20–24 h of recovery, the percentage of dead cells started to decrease while the OD began to increase (Fig 2C). We thus identified 24 hours as the inflection point, where each barcode is still unique, and live cells in the population have the ability to divide such that single cells can produce distinct, identifiable lineages.

One of the important prerequisites we set for the system was to be able to produce identical copies of libraries to allow comparisons between experiments. This was achieved using the following procedure. First, we sorted out the desired number of live cells at the time point we identified previously as the one at which single cells produce distinct lineages. Then we allowed the required amount of time for the cell population to expand sufficiently to produce 100 copies of each barcoded yeast (approximately 32 hours) and, finally, divided the culture into 100 aliquots and froze them in glycerol stock for the experiments. We note that the number of library copies produced can be changed by modulating the time the culture containing the sorted cells is allowed to expand and the number of aliquots accordingly. However, care must be taken to keep the expansion time short enough for the probability of acquiring mutations before the start of the experiment to be small. A detailed description and protocol of the transformation process can be found in the methods section.

**Validation of the barcoding and analysis procedure.** To assess the effect of various library preparation parameters on lineage makeup, we created seven mini-libraries that varied in the following parameters: the number of transformations performed (1 or 10), the number of cells sorted ($10\times10^3$, $55\times10^3$, $110\times10^3$) and the expansion times after transformation (31h or 34 h) (S1 Table). The compositions of the populations in the resulting mini-libraries were determined by deep-sequencing the barcodes of four random aliquots from each mini-library. The distribution of barcodes and the similarity between aliquots were unaffected by transformation parameters nor by cell sorting (S1 Table), demonstrating the robustness of the procedure. The number of viable lineages directly correlated with the number of cells sorted, although the precise number of lineages in the population was ten times lower than the number of lineages sorted. To maximize the number of barcodes in the population for further experiments, and given the above similarities, we pooled all mini-libraries into a larger one, which was then divided into 100 aliquots and frozen until used. To analyse the barcode composition of the resulting pooled library, we sequenced four of the aliquots. We found that these contain nearly identical barcode contents distributed log-normally, as shown in Fig 2D. Almost all 26,000 lineages were observed in all four samples (two were missing from one

sample), with lineage frequencies exhibiting high between-aliquot correlations (r = 0.990–0.999; $p < 0.001$).

Next, we wanted to verify that our experimental procedures (as described in Methods) produce an accurate description of the population. We, therefore, constructed two mock communities of known composition and subjected them to our analysis pipeline. To generate the mock populations, we first plated our library, picked eight colonies, and used generic primers spanning the barcode region to amplify and sequence (Sanger) the barcode of each colony. DNA was then extracted from the identified colonies and was mixed at two different ratios to construct two mock mixtures simulating different lineage proportions. Mock1 included equal proportions of all eight lineages, and Mock2 was composed of a non-uniform distribution, 50% of which is made up of lineage F and the remaining by equal proportions of the residual seven lineages. Fig 2E presents the expected population structures (bottom illustrations) along with the ones obtained by sequencing their barcodes on two separate occasions. The average error observed for the percentage of a lineage within a population is 1.8%, with lineage F showing the greatest error in Mock2 (7.5%). The fact that errors are almost identical when comparing the two different sequencing runs indicates that the variations from the expected structure are due to minor inaccuracies in the mock populations' construction process rather than sequencing biases and errors.

## Application of gUMI-BEAR in two experimental settings

**Tracking evolutionary dynamics.** We first tested our system in a microbial evolution experiment setting, which is an area in which cellular barcoding has provided considerable insights [23,24]. These types of experiments are usually carried out by serial dilution of a barcoded microbial population or in chemostats However, we wanted to retain as many barcodes as possible by reducing the elimination of low-frequency lineages by repeated bottlenecks, and thus we maintained the populations in turbidostats in a 100ml volume. In this setup, the cell density is maintained within a narrow window, in our case at an OD between 1.8–2.0, by diluting the cultures as soon as they reach the upper limit of the range until they reach the lower one. The cells reach exponential phase at the beginning of the experiment and are maintained in that growth state throughout, receiving rich media for the entirety of the experiment at a quantity that depends on their growth rate.

Cells were grown in the turbidostats in YPD media for 44 days while temperatures were varied three times to induce changes in the structure of the population. These were expected to occur due to differential fitness changes in individual lineages in the population [6,12,14,25,26]. We initially grew the population at 30°C for two days, raised the temperature to 39°C for eleven days, then to 41°C for 27 days, finally reducing it back to 39°C for the remaining four days of the experiment. Samples were collected throughout the experiment and sequenced to determine the clonal makeup of the population at each time point.

We tested the sequencing depth used in the analyses to rule out an inadequate one, which would make the results inaccurate, or oversampling, which would increase sampling costs unnecessarily. To this end, we constructed rarefaction curves for both the number of unique lineages detected and their distribution using an increasing number of reads at different time points along the experiment. As seen in Fig 3C and 3D, for all time points tested, both parameters plateau quickly at a low percentage of reads, indicating that our sequencing depth is sufficient. Since the distribution of lineage frequencies is more uniform at the early stages of the experiment (Fig 3D, inset), the number of reads at which it reaches the plateau increases with time but is still well below our sequencing depth of ~10 million reads per sample.

Using our analyses, we were able to follow a population that initially consisted of 26,000 distinct lineages (Fig 3C) distributed log-normally (Fig 3D, inset) and reveal frequency changes

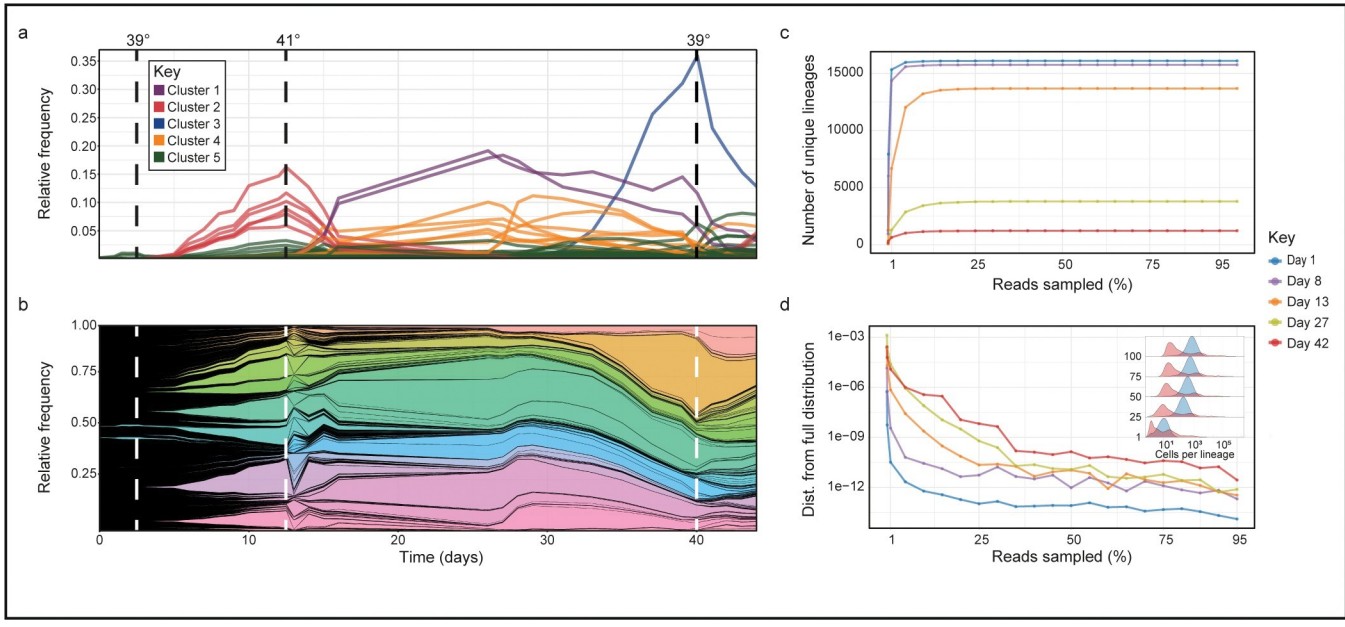

**Fig 3. Applying the gUMI-BEAR method to track evolutionary dynamics. (a, b)** Lineage trajectories throughout a 44-day experiment in which temperatures were altered (vertical dotted lines) to induce fitness fluctuations in the population. **(a)** Each lineage is represented by a line. Colours were assigned to each lineage based on K-means clustering performed on their trajectories throughout the experiment (as seen in the key). **(b)** Muller plot where each lineage is represented by a different colour, and the width of the coloured region at each timepoint is proportional to the lineage's relative frequency. **(c, d)** Rarefaction curves for the number of lineages **(c)** and lineage distributions **(d)** obtained by sampling an increasing number of reads at five time points (depicted by the different colours as defined in the key). **(d)** The accuracy of the distribution is quantified by the Wasserstein distance between the distribution obtained with a reduced number of reads and the original distribution (based on 100% of the reads). The inset shows a ridge plot of the frequency distribution of barcodes in the population quantified by sampling an increasing number of reads (y-axis, percentage of population sampled) from Day 1 (blue) and Day 44 (red).

of individual lineages. The size of the population dropped to 16,000 after the first day, after which the population underwent several cycles of clonal soft sweeps in which distinct lineages rose in frequency to become dominant (Fig 3A and 3B), with other lineages going down or remaining at low frequencies. The three temperature shifts induced sharp changes in population structure, the most dramatic manifestation of which was the immediate decline in abundance of the lineages that had been most successful under the previous condition, indicating that they had lost fitness relative to others. Surprisingly, this also occurred when we increased the temperature from 39°C to 41°C, implying that adaptation to high temperatures does not necessarily occur gradually. Notably, despite the high level of stress, ~2500 of the lineages (~15%) were not diluted out and remained in the population throughout the entire 44 days of the experiment (~128 generations). These results indicate that we were able to follow changes in the structure of a population consisting of a large number of barcodes at very low frequencies.

**Large scale tracking of gene variants.** Next, we wanted to show how gUMI-BEAR can be used for large-scale mutational screening, where each variant is tagged by a unique barcode, and its competitive success can be measured in terms of barcode abundance. We chose *Hsp82* to demonstrate the method's capability to screen genes that are longer than the NGS read length and because the protein it encodes affects fitness gain and loss via variability control through chaperone-like activity [27,28]. The number of possible *Hsp82* double mutants is over $2\times10^6$ and directed mutagenesis is unrealistic because of the work and cost involved in producing and testing each variant. However, we were essentially not limited in the number of variants we tested, which gave us freedom from the need to design specific mutants and allowed us to, instead, use a random mutagenesis kit (Genemorph II by Agilent part number

200550, see Materials and Methods). We performed mutagenesis on the *Hsp82* gene amplified from the *S. cerevisiae* BY4741 strain using the Genemorph kit (forward primer 100 bp upstream of the gene; reverse primer 100 bp downstream of the gene), with PCR parameters adjusted to produce 0–4.5 mutations per kb (annealing at 60˚C, 25 cycles, 2 min elongation time, and 90 ng genomic DNA template).

The resulting variants were conjugated to the gUMI-box, this time with the LHA and RHA complementary to the *Hsp82* gene. This created a donor molecule ready to be transformed into the native locus in the yeast genome, replacing the wild-type *Hsp82* (Fig 4B and Methods). Identical aliquots of the library were prepared in a manner similar to that described above, which allows exact repeats to be performed as well as to test the library of variants in other settings. Overall, we produced a library of 1,745,260 yeast clones, each carrying a randomly mutated and uniquely barcoded *Hsp82* variant.

The ability to test randomly mutated variants has the advantage of not introducing bias in their design. We wanted to test the effect of the assembly process on the distribution of mutations and, therefore deep-sequenced amplicons from two stages: after mutagenesis of the gene and after assembly of the final donor molecule containing the gUMI-box. A comparison of the sequences of the *Hsp82* variants emerging from the mutagenesis step and of the donor molecules revealed a decrease in the number of variants after integration of the ORF. This is to be expected, as the mutagenesis process is performed in a sequence agnostic manner, whereas genomic integration has biological constraints and, as such, eliminates some mutations. We then calculated the distribution of Shannon's information entropy of mutated nucleotides per site (see Fig 4A and S4 Fig). This distribution was identical to that of the molecules emerging from the Genemorph process showing that integration into the gUMI-box during the donor DNA assembly process does not introduce bias (Fig 4A). The mutated ORF within the donor molecules displayed the same entropy distribution in three independent assembly replicates, confirming that the process of integration into the gUMI-box was robust.

To allow variants to compete with each other, we used batch mini-cultures to grow four replicates at 30˚C, sampling and diluting at a rate of 1:100 every 12 hours, for a total of 120 hours. Samples were sequenced to yield the relative abundance of each lineage as a function of time (Fig 4D). We found the distribution of lineage frequencies to be heavily skewed throughout the experiment, with 50 lineages comprising almost all (99%) of the population. Of these, lineages in ranks 25–50 were each present at a relative abundance of 1% or less and all were successfully tracked by the gUMI-BEAR method. While the frequency of some lineages increased or decreased in the experiment, the abundance of most lineages remained constant, indicating relatively small fitness differences. The trajectory of the most successful lineages, namely those ranked among the top 50 in terms of their abundance, was reproducible between replicates, as indicated by the high correlations between different replicates (r = 0.89 ± 0.034) and the small standard deviation (Fig 4D). This signifies that the dynamics were determined by the variants that entered the experiment and not by adaptations that occurred throughout it, which could differ between repeats.

To demonstrate the capability of the system to identify variants of interest, we chose seven lineages for further analyses. As a first step, we used their reverse gUMI sequences as PCR primers to isolate specific variants. We were able to sequence a variant ranked as low as the 47[th] rank in a sample taken after 108 hours, constituting 0.19% of the population, demonstrating how even low-frequency lineages can be directly isolated from a mixed population (Fig 4D and 4E). The complete sequences of the *Hsp82* variants were then determined by Sanger sequencing using five primers distributed along the gene (Fig 4E). A comparison of the sequences obtained in this manner with sequences from full-length third-generation sequencing (Loop Genomics) confirmed the quality of variant sequencing (Fig 4E). Thus, with a single

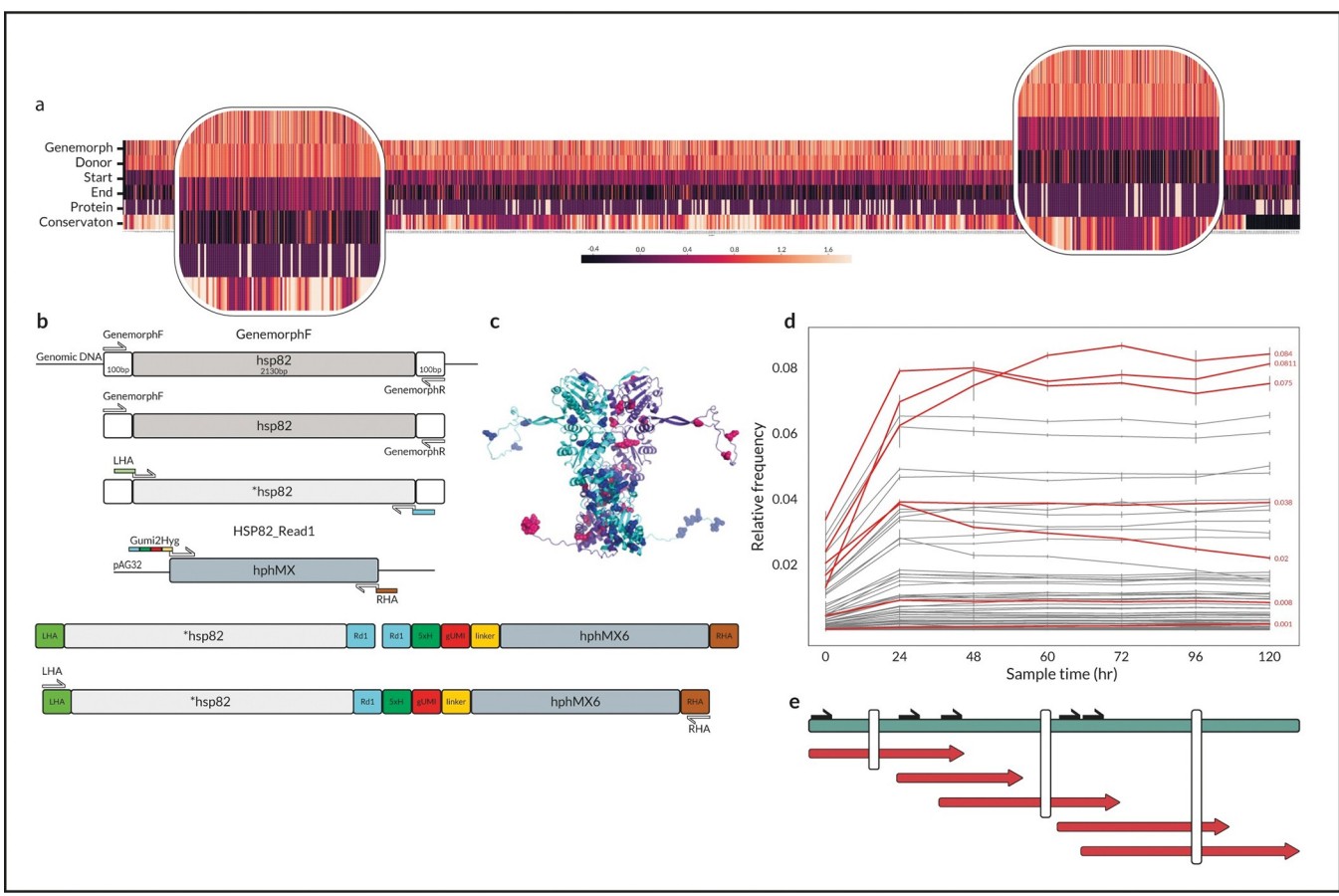

**Fig 4. Applying gUMI-BEAR to track gene variants. (a)** Heatmap showing entropy values (4 upper bands, as per the colour bar) for mutations in each nucleotide in the *Hsp82* variants for various steps in the library construction process. For each position, mutational proportions were normalised to sequencing depth, and Shannon's entropy was calculated (dark to bright represents low to high entropy). From top to bottom: The Genemorph panel refers to *Hsp82* variants created by insertion of random mutations using PCR with the Genemorph II kit. The Donor panel refers to the final donor molecule before transformation (mean entropy of three replicates). The panels labelled Start and End show the population at the start of the experiment, where cells have been transformed with the donor molecules and selected for successful transformants (mean of three replicates), and at the end of the experiment, respectively. The Protein panel shows mutational occurrence in the translated protein obtained by long-read sequencing of full-length variants (mutations are identified in white). The Conservation panel shows the conservation score for each position calculated using the ConSurf[18] server (dark for conserved and bright for variable regions). The insets show two zoomed-in regions exhibiting different patterns where both conserved and variable regions can have non-synonymous mutations. **(b)** Donor construction process as described in the Materials and Methods section. GenemorphF and GenemorphR are primers used to amplify the native *Hsp82* from the genome; LHA and RHA stand for Left/Right Homologous Arm and are added to enable correct transformation to the chosen locus; Gumy2Hyg is a forward primer containing all elements that are added to the donor molecule including the illumina read1 sequencing binding site (Rd1), a five non-guanine nucleotide sequence (5xH), the genomic unique molecular identifier (gUMI) and the linker sequence which is used for later preparation of the barcode molecule for sequencing. Briefly, *Hsp82* variants were produced by PCR amplification of the gene from the *S. cerevisiae* BY4741 strain using the Genmorph II kit. These were then conjugated to the gUMI-box, with LHA and RHA complementary to the *Hsp82* gene, which allowed them to replace the wild-type in the CRISPR/Cas9 reaction **(c)** The Hsp82 protein dimer (shown in green and purple). All mutations found by long-read sequencing are shown as red and cyan balls in the Hsp82 dimer. The image was produced with the Protein Homology/Analogy Recognition Engine (Phyre2). **(d)** Changes in relative lineage abundances within the yeast population over time. Each line represents the mean relative abundance of a lineage in the four replicates as a function of time, with error bars showing the standard deviation between replicates. Red lines mark the seven lineages whose *Hsp82* variant was amplified using the gUMI sequence as a reverse primer and then Sanger-sequenced. **(e)** A scheme presenting Sanger sequencing using five forward generic primers (black arrows) distributed along the *Hsp82* ORF with a specific gUMI reverse complementary primer that was used to obtain a constructed full-length sequence of the variant (Red). The constructed variants are compared to the full-length sequence of the variant achieved by long-read sequencing (Loop Genomics) in green (Identified SNPs are indicated by the white lines).

PCR reaction, performed on the entire population, and five Sanger sequencings, we were able to distinguish a variant that, using present methods, would require the sequencing of 500 single-cell colonies, *on average*. Having identified variants of interest, one can go on to test them in isolation by inserting the relevant ORF into wild-type strains.

When we compared short reads obtained from the initial yeast library, after transformation and expansion, with the long reads from the population after it had undergone 120 hours of evolution (Fig 4A), there was an overall decline in entropy, indicating that the experiment began with numerous mutations holding reduced fitness which were eliminated as growth continued. In the final library, we also found regions where no mutations had occurred, presumably because mutations in these areas reduce fitness significantly. The process of growing a population that harbours random mutations in a gene thus provided a biological filter, at evolutionarily relevant conditions, that eliminated lethal or function-harming mutations while allowing the propagation of mutations that were neutral or that provided a fitness advantage.

## Discussion

In this work, we describe a single-cell barcoding method that is versatile, easy to implement, and cost-effective, thus lending itself to a host of experimental scenarios. These traits are achieved due to a combination of features: Firstly, we used CRISPR/cas9 coupled with the cellular HDR to integrate the barcode containing gUMI-box, thereby achieving a high yield of transformants which in turn allows the construction of libraries of extremely large size. Furthermore, the gUMI-box integration site can easily be changed by simply designing a new gRNA and a corresponding donor DNA. Secondly, we perform a sorting step following recovery from the transformation to obtain control over the number of lineages participating in the experiment. Importantly, our method allows the assembly of multiple identical replicates of the same initial population, opening the way for performing precise repeats and for comparisons between the behaviour of the same populations of cells under different experimental scenarios. This, in turn, makes it possible to apply barcoding towards investigating the role of randomness and determinism in highly complex biological systems [22,29,30].

The gUMI-box, together with the simple and efficient protocol we applied to prepare gUMI-barcoded sequences for deep-sequencing (Fig 1G), eliminates the need for multiple preparation steps and, at the same time, provides high resolution and accuracy and supports the use of small samples.

We presented two applications to demonstrate the potential and adaptability of the gUMI-BEAR method. The first application exhibited its scale and precision in tracking ~16,000 lineages through 30 time points over a period of 44 days and revealed detailed evolutionary dynamics resulting from the complex environmental regime. We obtained an intricate picture of the rich population dynamics that arise as multiple lineages adopt numerous modes of exploration and exhibit varying adaptabilities to changing environmental conditions. Attaining such a picture through standard population studies, such as bulk RNA sequencing, whole-genome sequencing, or plating assays, would be financially and/or analytically impractical.

In the second experiment, we used gUMI-BEAR to explore gene-variant fitness and dynamics of populations with induced fitness variations. By coupling a gUMI to each variant, we were able to scan a large number of random mutations produced *in vitro* in an evolutionarily relevant context in which they compete with each other, eliminating the need for either fitness testing of monoclonal variants or costly in-depth sequencing. Due to our ability to produce identical libraries, we were able to observe largely deterministic dynamics for this population. Importantly, we used reverse-gUMI sequences as PCR primers to target specific variants and then used Sanger sequencing to identify the variation without the need for laborious work to single out variants and expensive long-read sequencing, thereby alleviating the previous limitation on the complexity of variants that could be tested in parallel methods.

Our method thus allows in-depth exploration of millions of variants of any complexity, all living and competing as a population, while also providing the ability to explore interesting

variants, even at very low frequency, using simple techniques such as PCR and Sanger sequencing.

We note that the gUMI-BEAR method can be utilised for variant screening in more complex systems, consisting of multiple genes, by adding a short sequence to identify each ORF. Under such a configuration, the relevant genes can be multiplexed to create an array of randomly mutated genes interacting with each other to enable the study of complex enzymatic pathways.

Using our ability to produce identical libraries, one can test the dynamics of the same populations or variants under different treatments. This is relevant as fitness may depend on environmental factors. Thus, our method allows studying gene-environment relations on a large scale.

Additional settings where the gUMI-BEAR method can be implemented are the Pharma and Biotech industries, particularly during the scale-up of fermentation processes and during the production phase. Specifically, it could be used to identify deleterious processes before they affect production or to single out the key events that impair large-scale production and find ways to overcome them [31].

We designed gUMI-BEAR to be an easy-to-use and robust method to track changes in population makeup and variant fitness. Previously available methods [1,3,10] to track such changes were not scalable and were difficult to deploy on new lines and modalities or more costly. A library can be created using the gUMI-BEAR method in a few days and our results show that it provides an accurate, cost-effective, and versatile means of tracking subtle changes that propagate into major population shifts in a way that can easily be adapted to other biological systems.

## Materials and methods

The methods described below were common to both of the experiments conducted, except where otherwise noted.

### Construct assembly

**gUMI-box (Genomic Unique Molecular Identifier).**  The gUMI-box contains a 24 bp random sequence that serves as a transmissible barcode for each lineage in the population. The gUMI is flanked by the read 1 Illumina generic sequence on one side and a 15 bp generic sequence called a linker on the other side. The linker is used as a target sequence for DNA amplification in the deep-sequencing library preparation step, such that two-step PCR produces a library containing all elements necessary for sequencing on Illumina platforms (Fig 1G). An additional hygromycin-resistance cassette (HGmx6; obtained from the vector pAG32 [32]) is added downstream of the linker sequence to allow selection and to prevent contamination (Fig 2A). As a final step, two homologous arms are added to the gUMI-box upstream and downstream of the genomic integration site. All elements are combined into one linear DNA construct, the gUMI-box, which is then transformed into a population of yeast cells.

**Donor construction for tracking evolutionary dynamics.**  Donor DNA was built using two primers containing 50 bp sequences inserted upstream (left homologous arm; LHA) and downstream (right homologous arm; RHA) of the integration site in the *YBR209W* locus, which encodes a putative protein of unknown function and a non-essential gene [33]. The forward primer consists of the specific LHA and most components of the gUMI-box. At its 3' end, it contains a complementary sequence to the HGmx6 cassette from pAG32. The reverse primer consists of the specific RHA and a sequence complementary to the end of the HGmx6 cassette. PCR using these two primers produces the full linear DNA used for transformations

in later steps (Fig 2A). The double-strand break site was determined using CHOPCHOP [22]. The pCAS plasmid was cloned by restriction-free (RF) cloning with two primers containing the gRNA sequence (5'-TAGAGCGTCAATCAAGAAAG-3') as described previously[13].

**Donor construction for large scale tracking of gene variants.** Six-step PCR was utilized to obtain a full-length donor DNA comprising 5'-LHA-d*Hsp82*-gUMI-box-RHA-3'. *HSP82* was amplified from *S. cerevisiae* BY4741 gDNA to obtain a wild-type copy of the gene. Random mutations were introduced using the Genemorph II random mutagenesis kit (Agilent, 200550). To achieve a low mutation rate (0–4.5 mutations/kb), 1 μg of the template DNA was amplified using this kit in a 30-cycle PCR protocol. A series of PCRs containing the gUMI-box and the *Hsp82 were performed to create the full-length construct (Fig 4B and Supplementary methods).

Two pCAS plasmids were cloned by RF cloning, each with two primers containing the gRNA sequence for the cleavage site upstream (5'-CAAACAAACACGCAAAGATA-3') and downstream of the *Hsp82* gene (5'-AGCTGACACCGAAATGGAAG-3').

## Transformation

**For tracking evolutionary dynamics.** Competent cells of *S. cerevisiae* BY4741 were prepared and each resulting aliquot was transformed with donor DNA (5 μg) and cloned pCAS (2 μg) using the protocol described by Ryan et al [22,34]. Following a three-hour recovery in 450 μL of YPD media at 30°C, aliquots were mixed and incubated for 20 h in 100 mL YPD broth with hygromycin B (300 μg/mL) at 30°C at 220 rpm. To accurately determine the number of lineages involved in each experiment, following the 20 hr recovery phase, the cells were stained with propidium iodide[27] to mark perforated cells and the desired number of viable cells was sorted. The library was grown in 100 mL YPD with hygromycin B and was allowed to propagate for 32 h (~16 generations). This step was followed by the division of the library into 100 identical aliquots that were then immediately frozen in 50% Glycerol at -80°C (Fig 1A). To examine the effects of several parameters on the barcoded library structure we performed the abovementioned process seven times, varying the number of competent cells used to build each library, the time allowed for recovery following transformation, total number of cells sorted following recovery, and expansion time following recovery. S1 Table presents the parameters used in the seven libraries we built.

**For large scale tracking of gene variants.** Cells were transformed as described above. Following recovery, three aliquots were incubated for 20 h in 10 mL YPD with hygromycin B at 30°C at 220 rpm to select transformants. The library was diluted 100-fold and was allowed to propagate for an additional 30 h. This step was followed by the division of the library into 100 identical aliquots that were immediately frozen in 50% Glycerol at -80°C.

## Experimental outline

**Tracking evolutionary dynamics.** The experiment was performed in a turbidostat (Eppendorf DASbox Mini Bioreactor System) at a constant volume of 100 mL YPD supplemented with hygromycin B (200 μg/mL). Genetic drift was kept to a minimum by maintaining the concentration of the culture within a narrow range of 0.9–1.1 (AU), eliminating large dilution steps. Cells were grown for a total of 44 days The initial temperature of the culture was set to 30°C. After 48 hours, the temperature was increased to 39°C for eleven days, 41°C for 27 days and reduced back to 39°C for the remainder of the experiment (Fig 3A and 3B). A total of 31 samples were taken throughout the experiment in three replicates, for DNA and RNA extraction and glycerol stock of viable cells. To prepare the DNA/RNA samples for later analysis we centrifuged each sample at 8.6g for 2 minutes, removed the supernatant, and flash frozen

them in liquid nitrogen at -80˚C. Glycerol samples were treated similarly, except that following supernatant removal, the cells were resuspended in YPD with 25% Glycerol.

**Large scale tracking of gene variants.** Four library aliquots were mixed to create a homogeneous population and to minimise minor differences between aliquots. The mix was evenly distributed to four replicates that were grown in 100 mL Erlenmeyer flasks containing 25 mL YPD with hygromycin B (200 µg/mL). Environmental conditions were maintained at a constant temperature of 30˚C and constant mixing at 220 rpm on a magnetic stirrer. The experiment spanned five days such that every 12 hours, as cells approached the end of their exponential growth phase, a sample was taken, and the culture was diluted 1:100.

## Sample preparation and sequencing

DNA was extracted using the MasterPure Yeast DNA purification kit (Epicentre, MPY80200), and concentrations were measured by Qubit. A sub-sample (150 ng) was taken from each sample to serve as a template for four-cycle PCR in which the barcode region was linearly amplified, and the internal index and UMI were added (Fig 1G). Five samples from this reaction were pooled on a Qiagen PCR clean-up column (Cat. 28104) and eluted in a 50 µL elution buffer. These samples were further cleaned by 1.8× Ampure XP beads (Cat. A63881). From each pool, 10 µL were used for a 12–17 cycles PCR using Illumina indexed primers. Samples were purified using 1.8× Ampure XP beads, and their quality was evaluated using Agilent Tape-station (Cat. 5067–5584) and Qubit concentration measurements (Cat. Q32853). All samples were diluted to a concentration of 0.5 ng/µL, and 10 µL of each diluted sample was used to create the final library. This sample was further diluted to a final concentration of 2 nM in a final volume of 100 µL. The library was sequenced at the Weizmann Institute of Science's G-INCPM unit on an Illumina NovaSeq platform with the following cycles: 29|10|10|26.

## Preliminary analysis of raw data

Raw data were filtered to remove low-quality reads. Demultiplexing of samples was performed in two parts using Illumina indices and the internal indices. In order to count lineage frequency and determine which barcodes originated from a cell and which from sequencing mistakes, the CD-HIT [35,36] algorithm was used in two steps. In the first, seed sequences are created from samples taken throughout the experiment. Next, sequences from each specific time point were compared with the seed sequences list (using the cd-hit-est2d function). In each step, we allowed up to a two bp mismatch before assigning the sequence to a new seed. The CD-HIT function clst2txt.pl was used to create lineage count data-frames for each sample. The final output of this process contained many clusters of sequences that were each classified as a single lineage for further analysis. In the final step of the preliminary analysis, custom, in-house R-code was used to pool together all sequences from the same cluster with the same UMI to one cell. The final output of the abovementioned pipeline accurately measured the number of actual, single cells originating from a specific lineage for every time point after correction of PCR biases. For full script see rezenman/gumi_bear_code github page.

## Supporting information

**S1 Fig. Nucleotide composition by position for the gUMI from donor DNA and lineages.** gUMI region of the donor DNA prior to transformation (upper panel) and DNA from lineages participating in the experiment (lower panel) were extracted and deep-sequenced to reveal the nucleotide composition for each position in the gUMI (x-axis). The overall height of each base is proportional to its probability (y-axis). Colours represent different nucleotides. (DOCX)

**S2 Fig. Guanine-Cytosine content distributions for the gUMI from three stages in the experiment with no initial fitness variations.** GC content was measured based on sequences derived from all lineages at three steps during library construction. (left, red) Donor DNA prior to transformation, (middle, green) from lineages at the beginning of the experiment, (right, blue) from lineages at the end of the experiment. All data are presented in histograms to reveal the GC content distribution in the populations.
(DOCX)

**S3 Fig. Mean doubling time and glucose concentration for the experiment in population exhibiting no initial fitness variations.** Doubling time for the whole culture was calculated based on dilution rate of the turbidostats and is presented on a log scale (left y-axis, orange line) and plotted as function of time. Glucose concentrations were measured for each sample taken (right y-axis, blue line). The experiment started at an optimal growth temperature of 30˚C that was altered to induce stress as noted by the dotted lines.
(DOCX)

**S4 Fig. Entropy kernel density estimations for the *HSP82* ORF before and after incorporation into the gUMI-box.** The kernel density estimations of the Shannon entropy for nucleotide substitutions in the *HSP82* ORF in the genenmorph amplicon after *in vitro* mutagenesis (black) and three replicates of the assembled donor DNA (green, red and blue) are shown. Three peaks can be observed—first at entropy 0, in which all substations were to the same nucleotide, second at entropy 1, in which substitutions were equal between one of two possible substitutions, and finally, the third peak at entropy 1.5 in which an equal probability for all mutations is observed. The donor DNA distribution mimics the one produced by the genemorph, indicating that no bias was induced during donor construction.
(DOCX)

**S1 Table. Mini-library parameters.** Seven different mini-libraries were prepared while varying the following parameters: No. transformations—the factor applied to the volumes given in the protocol published by Ryan et al.[1], namely, 1 μg pCAS, 5 μg donor DNA, and 90 μL competent cells. Hence, No. of transformations = 10 indicates the use of 10 μg pCAS, 50 μg donor DNA, and 900 μL competent cells; $T_{sel}$, selection time, being the number of hours the library was grown for after transformation and before sorting; No. cells sorted, total number (in thousands) of viable cells sorted following staining with propidium iodide and selection (see Methods); $T_{exp}$, expansion time, being the time (hours) the library was allowed to expand before division into 100 aliquots; No. unique lineages, number of viable lineages in each mini-library as determined by deep-sequencing.
(DOCX)

**S1 File.**
(DOCX)

## Acknowledgments

We thank Shira Holand for the graphic design, Yonatan Nutkewitz for helpful discussions and help with experiments and Hadas Keren-Shaul for help with next-generation sequencing.

## Author Contributions

**Conceptualization:** Shahar Rezenman, Maor Knafo, Shiri Barad, Ziv Reich, Ruti Kapon.

**Data curation:** Shahar Rezenman, Maor Knafo, Ivgeni Tsigalnitski, Ghil Jona, Dikla Levi.

**Formal analysis:** Shahar Rezenman, Maor Knafo, Ivgeni Tsigalnitski, Orly Dym.

**Funding acquisition:** Ziv Reich.

**Investigation:** Shahar Rezenman, Maor Knafo, Ivgeni Tsigalnitski, Shiri Barad, Ghil Jona, Dikla Levi.

**Methodology:** Shahar Rezenman, Maor Knafo, Ivgeni Tsigalnitski, Shiri Barad, Ghil Jona, Dikla Levi, Ziv Reich, Ruti Kapon.

**Project administration:** Ruti Kapon.

**Software:** Shahar Rezenman, Maor Knafo, Ivgeni Tsigalnitski, Shiri Barad.

**Supervision:** Ziv Reich, Ruti Kapon.

**Validation:** Shahar Rezenman, Maor Knafo, Ivgeni Tsigalnitski, Shiri Barad, Ziv Reich, Ruti Kapon.

**Visualization:** Shahar Rezenman, Maor Knafo, Ivgeni Tsigalnitski, Orly Dym.

**Writing – original draft:** Shahar Rezenman, Maor Knafo, Ivgeni Tsigalnitski, Shiri Barad.

**Writing – review & editing:** Shahar Rezenman, Maor Knafo, Ivgeni Tsigalnitski, Shiri Barad, Ghil Jona, Orly Dym, Ziv Reich, Ruti Kapon.

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
