## [Decision Letter · Decision Letter 0]

19 Apr 2023

PONE-D-23-06106gUMI-BEAR, a modular, unsupervised population barcoding method to track variants and evolution at high resolutionPLOS ONE

Dear Dr. Kapon,

Thank you for submitting your manuscript to PLOS ONE. After careful consideration, we feel that it has merit but does not fully meet PLOS ONE’s publication criteria as it currently stands. Therefore, we invite you to submit a revised version of the manuscript that addresses the points raised during the review process.

We look forward to receiving your revised manuscript.

Kind regards,

Sudhir Kumar Rai, Ph.D

Academic Editor

PLOS ONE

Journal Requirements:

2. We noted in your submission details that a portion of your manuscript may have been presented or published elsewhere. [a previous version of this manuscript has been uploaded to the BioRxive : bioRxiv 2022.09.01.506035; doi: https://doi.org/10.1101/2022.09.01.506035] Please clarify whether this [conference proceeding or publication] was peer-reviewed and formally published. If this work was previously peer-reviewed and published, in the cover letter please provide the reason that this work does not constitute dual publication and should be included in the current manuscript.

4. Please upload a copy of Supporting Information Figure/Table/etc. Supplementary Figure 1 to 5 and Supplementary Table 1 which you refer to in your text on page 31.

Reviewers' comments:

Reviewer's Responses to Questions

**Comments to the Author**

1. Is the manuscript technically sound, and do the data support the conclusions?

Reviewer #1: Yes

Reviewer #2: Yes

2. Has the statistical analysis been performed appropriately and rigorously? 

Reviewer #1: Yes

Reviewer #2: Yes

3. Have the authors made all data underlying the findings in their manuscript fully available?

Reviewer #1: Yes

Reviewer #2: Yes

4. Is the manuscript presented in an intelligible fashion and written in standard English?

Reviewer #1: Yes

Reviewer #2: Yes

5. Review Comments to the Author

Reviewer #1: I have reviewed the paper titled "gUMI-BEAR, a modular, unsupervised population barcoding method to track variants and evolution at high resolution" and I find it well-written and carefully executed study.

The authors have presented a versatile and cost-effective method for generating and screening libraries of yeast mutants using CRISPR/Cas9 and homology-directed repair to introduce a unique molecular barcode (gUMI-box) into the yeast genome. The study demonstrates that the method can generate large libraries of yeast mutants and that the resulting populations can be tracked and analyzed using the gUMI-BEAR method. The authors have also shown that the system can identify biologically relevant mutations and variants of interest.

However, I have some minor comments for the authors which I believe would improve the paper:

1-The purpose of using homology model of HSP82 is not very clear to me. Additionally, I was unable to locate the discussion for Fig. 4c and Supplementary Figure 5 in the text.

2- I suggest that the authors consider using the HSP82 model from Alphafold2 as it would improve the accuracy of their protein-protein interaction models.

3- The authors could elaborate on how they generated the protein-protein interaction models of HSP82 with Cdc37 and Cdk4.

4- It would be informative for readers if the authors could provide information on the HSP82 mutant highlighted in the homology model in table format, along with its consequences if known.

5- In lines 450-470, the authors describe that the gUMI-BEAR method can be used for variant screening in complex systems with multiple genes by adding a short sequence to identify each ORF. I suggest that the authors provide a few examples of complex enzymatic pathways to illustrate this.

Overall, I believe that this study provides a valuable contribution to the field and I recommend its publication in PLOS ONE after the authors have addressed the minor comments above.

Reviewer #2: The author introduces a novel method for tracking variant populations using gUMI-BEAR (genomic Unique Molecular Identifier Barcoded Enriched Associated Regions), a modular system with promising applications in variant tracking.

The method described in the paper is technically well-explained, and the author provides two examples of its use in tracking evolutionary dynamics and gene variants. I believe that these examples can be replicated in other labs. However, it is worth noting that there are some minor typographical errors in the paper which needs to be corrected.

Typographical mistakes

Page line 237: Full stop missing.

Page 24 line 551: "analysis we centrifuged each sample at 8.6g for 2 minutes" define speed clearly.

Additionally, the paper would benefit from a more comprehensive comparative analysis with existing methods. Despite these limitations, I believe that this work is a valuable contribution to the field, and I recommend it for your consideration.

6. PLOS authors have the option to publish the peer review history of their article (what does this mean?). If published, this will include your full peer review and any attached files.

Reviewer #1: **Yes: **Badri N. Dubey

Reviewer #2: No

---

## [Author Response · Author response to Decision Letter 0]

15 May 2023

Reviewer #1:

We thank the reviewer for his helpful comments and suggestions. Below please find a detailed response to each point raised, along with the changes we made in the text to address it. 

1-The purpose of using homology model of HSP82 is not very clear to me. Additionally, I was unable to locate the discussion for Fig. 4c and Supplementary Figure 5 in the text.

• Following the reviewer’s suggestion, we have added the following explanation for the use of the homology model in the main text (page 18, rows 399-410): “To gain insights into the structural role of the different Hsp82 mutations, we utilise the deep learning algorithms implemented in the AlphaFold2 software (Google/DeepMind’s artificial intelligence (AI) system). The AlphaFold2-multimer was used to predict the Hsp82 homodimer of 28 different constructs from the highest ranking variants, each containing between one to four different mutations and some with deletion of the C-terminal segment (701ADTEMEEVD709). The WT Hsp82 homodimer model (residues 1-709) is shown in cyan and purple, and all mutations identified in all the variants are shown as blue and pink balls for the two Hsp82 monomers, respectively (Fig 4c). Most mutations do not involve the dimer interface and are unlikely to explain the relatively higher fitness of successful lineages. This demonstrates the utility of applying our system, together with random mutagenesis, to reveal variants of interest whose advantage could not have been predicted. “

• Reference to Figure 4c was made in the added text above (row 406) in the main text file). 

• We thank the reviewer for pointing out that the analysis presented in Figure 5 of the supplementary is beyond the scope of this text, and we have therefore omitted it.

2- I suggest that the authors consider using the HSP82 model from Alphafold2 as it would improve the accuracy of their protein-protein interaction models.

• As the reviewer suggested, we implemented the Alphafold2 model on 28 sequenced variants and found several mutations on the flexible residues that Phyre2 lacked the resolution to assign. We have replaced the model in Fig 4c with the new one and edited the figure caption (rows 341-345) to reflect the change. The caption to Fig 4c now reads: “Cartoon representation of the Hsp82 protein. The AlphaFold2-multimer was utilised to predict the structural model of the Hsp82 homodimer shown in cyan and purple. All mutations found by long-read sequencing are shown as blue and pink balls in the Hsp82 dimer. The image was produced using the PyMOL software package (Schrödinger, L. & DeLano, W., 2020 PyMOL).

• We added a paragraph describing the implementation of AlphaFold2 in the text on page 18, lines 399-410 that also addresses this (see point 1 above)

• We have also revised the description of the model in the second paragraph of the Supplementary results (page 7) to depict the new model obtained using AlphaFold2. The new description reads: “The crystal structure of the Hsp82 homodimer from Saccharomyces cerevisiae is known (PDB-ID 2CG9). The Hsp82 structure is missing two segments 217V-K261 and the C-terminal 678I-D709. To gain insights into the structural role of the different Hsp82 mutations we utilize the deep learning algorithms implemented in the AlphaFold2 software (Google/DeepMind’s artificial intelligence system2). The AlphaFold2-multimer was used to predict the Hsp82 homodimer of all 28 different constructs, obtained by Loop genomics long read sequencing, each containing between 1 to 4 different mutations and some with deletion of the C-terminal segment (701ADTEMEEVD709). The different constructs contain 50 different mutations (I20T, A87T, L127I, N164K, V201M, D236E, E250D, K258I, K337E, K342, D352Y, E353V, N377I, N405I, A438V, Y445H, K449M, D459N, K480Q, I505V, K514E, K514R, T525A, D534Y, P549S, V566G, A574T, R591K, M593V, A595V, L598M, M603I, K618E, K626E, K640R, F664Y, V699I , V699D, P700D, A701, D702E, T703S, E704D, M705K, E706, E707, V708, and D709). The WT Hsp82 homodimer model (residues 1-709) is shown in cyan and purple, and all mutations found by long-read sequencing are shown as blue and pink balls for the two Hsp82 monomers, respectively (Fig 4c). The dimer interface of the various constructs of the Hsp82 model contains between 130-260 contacts up to 3.5 Å involving 60-80 different amino acid residues. Approximately half of them appear in the dimer interface of each of the 28 different constructs and 8 are involved in salt bridge contacts (E4-K86, R380-E381, K423-D600, and K484-E660). Only 4 of the residues involved in the dimer interface are mutated residues found in the following constructs; I505V (Hsp-11), A595V (Hsp-4), M603I (Hsp-20), and L598M (Hsp-29). Since the dimer interface of each of the Hsp82 constructs involves between 60-80 different residues, it seems reasonable to assume that a single point mutation in a residue involved in the dimer interface will not prevent dimer formation.”

 

3- The authors could elaborate on how they generated the protein-protein interaction models of HSP82 with Cdc37 and Cdk4.

Following the reviewer’s first comment above we omitted the figure that showed these models (see reply to comment 1)

4- It would be informative for readers if the authors could provide information on the HSP82 mutant highlighted in the homology model in table format, along with its consequences if known.

We agree with the reviewer about the importance of such data, but we could not find an explanation for the advantage provided by the mutations. We believe this demonstrates the strength of the method in revealing variants whose advantage could not have been predicted based on known structures. The last line in the paragraph added in response to the first comment above also addresses this point.

“To gain insights into the structural role of the different Hsp82 mutations, we utilise the deep learning algorithms implemented in the AlphaFold2 software (Google/DeepMind’s artificial intelligence (AI) system). The AlphaFold2-multimer was used to predict the Hsp82 homodimer of 28 different constructs from the highest ranking variants, each containing between one to four different mutations and some with deletion of the C-terminal segment (701ADTEMEEVD709). The WT Hsp82 homodimer model (residues 1-709) is shown in cyan and purple, and all mutations identified in all the variants are shown as blue and pink balls for the two Hsp82 monomers, respectively (Fig 4c). Most mutations do not involve the dimer interface and are unlikely to explain the relatively higher fitness of successful lineages. This demonstrates the utility of applying our system, together with random mutagenesis, to reveal variants of interest whose advantage could not have been predicted. 

5- In lines 450-470, the authors describe that the gUMI-BEAR method can be used for variant screening in complex systems with multiple genes by adding a short sequence to identify each ORF. I suggest that the authors provide a few examples of complex enzymatic pathways to illustrate this.

We have added a few sentences at the end of the paragraph that mentions complex enzymatic pathways to illustrate such an application. Specifically, we have added the following to the text (page 21, lines 469-474 ):

“For example, in order to optimise the glycolysis pathway, one could produce randomly generated, gUMI-barcoded variants of each of the enzymes that participate in it and follow the fitness of yeast harbouring different combinations of variants thus optimising the entire pathway. The system also has the potential to optimise industrial and medicinal production pathways by, for example, improving protein secretory pathways to increase recombinant protein expression.”

Reviewer #2

Page line 237: Full stop missing.

We thank the reviewer and have added a full stop on page 11, line 237 of the revised manuscript

Page 24 line 551: "analysis we centrifuged each sample at 8.6g for 2 minutes" define speed clearly.

We thank the reviewer for the comment and have added the relevant numbers to the text on page 25, line 570 which now reads “To prepare the DNA/RNA samples for later analysis we centrifuged each sample at 8600g (9500 RPM) for 2 minutes, removed the supernatant, and flash frozen them in liquid nitrogen at -80°C.”

Minor stylistic changes:

Page 18, line 397: For clarity, we modified the sentence 

“Having identified variants of interest, one can go on to test them in isolation by inserting the relevant ORF into wild-type strains.” and it now reads: 

“We note that identifying variants of interest allows one to go on to test them in isolation by inserting the relevant ORF into wild-type strains. “

 

Supplementary Figure 1 caption:

The beginning of the first line in the caption to Supplementary Figure 1 was changed from:

“Donor DNA prior to transformation ….” 

to:

“gUMI region of the donor DNA prior to transformation …”

Supplementary Figure 2 caption:

We edited the caption slightly from:

“GC content was measured based on sequences derived from all lineages at three steps during library construction. (Left, red) Donor DNA prior to transformation. (Middle, green) Lineages at the beginning of the experiment. (Right, blue) Lineages at the end of the experiment. All data are presented in histograms to reveal the GC content distribution in the populations.”

to:

“GC content was measured based on sequences derived from all lineages at three steps during library construction. (left, red) Donor DNA prior to transformation, (middle, green) from lineages at the beginning of the experiment, (right, blue) from lineages at the end of the experiment. All data are presented in histograms to reveal the GC content distribution in the populations.” 

Supplementary results page 7:

The title was edited to concur with S. Cerevisiae protein nomenclature from:

Proteomics analysis of hsp82 variants

to :

Proteomics analysis of Hsp82 variants

A spelling mistake in the second line of the second paragraph was corrected.

Supplementary methods, page 8, First paragraph on Restriction Free cloning :

We corrected the reference of CHOP-CHOP to reference 3. 

First line in last paragraph on page 8:

We replaced “For use with tracking” to “For use in tracking”.

Page 10, title below All primer sequences used for the donor construction process 

The title was changed from “No initial fitness differences experiment” to “Tracking evolutionary dynamics experiment” to match the name of the experiment in the main text.

Throughout the supplementary methods:

All “hsp82 gene” were replaced with “HSP82 gene” to concur with S. Cerevisiae nomenclature

---

## [Editor Report · Decision Letter 1]

22 May 2023

gUMI-BEAR, a modular, unsupervised population barcoding method to track variants and evolution at high resolution

PONE-D-23-06106R1

Dear Dr. Ruti Kapon,

We’re pleased to inform you that your manuscript has been judged scientifically suitable for publication and will be formally accepted for publication once it meets all outstanding technical requirements.

Kind regards,

Sudhir Kumar Rai, Ph.D

Academic Editor

PLOS ONE

---

## [Editor Report · Acceptance letter]

25 May 2023

PONE-D-23-06106R1 

gUMI-BEAR, a modular, unsupervised population barcoding method to track variants and evolution at high resolution 

Dear Dr. Kapon:

I'm pleased to inform you that your manuscript has been deemed suitable for publication in PLOS ONE. Congratulations! Your manuscript is now with our production department. 

Kind regards, 

on behalf of

Dr. Sudhir Kumar Rai 

Academic Editor

PLOS ONE